# Towards Hybrid-grained Feature Interaction Selection for Deep Sparse Network

**Fuyuan Lyu**[1]**, Xing Tang**[2][†]**, Dugang Liu**[4]**, Chen Ma**[3]**,**
**Weihong Luo**[2]**, Liang Chen**[2]**, Xiuqiang He**[2]**, Xue Liu**[1]
[1]McGill University, [2]FiT, Tencent, [3]City University of Hong Kong,
[4]Guangdong Laboratory of Artificial Intelligence and Digital Economy (SZ)
[†] corresponding author
fuyuan.lyu@mail.mcgill.ca, xing.tang@hotmail.com,
dugang.ldg@gmail.com, chenma@cityu.edu.hk,
{lobbyluo,leocchen,xiuqianghe}@tencent.com,
xue.liu@cs.mcgill.ca

## Abstract

Deep sparse networks are widely investigated as a neural network architecture for prediction tasks with high-dimensional sparse features, with which feature interaction selection is a critical component. While previous methods primarily focus on how to search feature interaction in a coarse-grained space, less attention has been given to a finer granularity. In this work, we introduce a hybrid-grained feature interaction selection approach that targets both feature field and feature value for deep sparse networks. To explore such expansive space, we propose a decomposed space which is calculated on the fly. We then develop a selection algorithm called OptFeature, which efficiently selects the feature interaction from both the feature field and the feature value simultaneously. Results from experiments on three large real-world benchmark datasets demonstrate that OptFeature performs well in terms of accuracy and efficiency. Additional studies support the feasibility of our method. All source code are publicly available[1].

## 1   Introduction

Deep Sparse Networks (DSNs) are commonly utilized neural network architectures for prediction tasks, designed to handle sparse and high-dimensional categorical features as inputs. These networks find widespread application in real-world scenarios such as advertisement recommendation, fraud detection, and more. For instance, in the context of advertisement recommendation, the input often comprises high-dimensional features like *user id* and *City*, which significantly contribute to the final prediction.

As depicted in Figure 1(a), the general DSN architecture consists of three components. Firstly, the embedding layer transforms different feature values into dense embeddings. Following this, feature interaction layers create feature interactions [4] based on the embeddings of raw features, as illustrated in Figure 1(b). Finally, the predictor makes the final prediction based on the features and their interactions. A core challenge in making accurate DSNs predictions is effectively capturing suitable feature interactions among input features [8, 11, 16, 15].

Various methods have been proposed to address the issue of modelling feature interactions. Wide&Deep first models human-crafted feature interactions using linear regression [22]. To eliminate the need for human expertise, DeepFM [8] models all second-order feature interactions by utilizing

---

[1]https://github.com/fuyuanlyu/OptFeature

a factorization machine [21] and adopts a multi-layer perceptron (MLP) as a predictor. Further advancements have replaced the factorization machine with different operations, such as product operations [18], cross network [24, 25], or an MLP component [19]. However, these methods have inherent drawbacks as they directly model all possible feature interactions. This inevitably introduces noise into the models, increasing their complexity and potentially degrading performance.

Neural architecture search(NAS) [12] has been introduced as a powerful approach for feature interaction selection in DSNs for both efficiency and effectiveness [11, 7, 15]. AutoFIS [11] first propose to select feature interactions by adding an attention gate to each possible one. PROFIT [7] suggests progressively selecting from a distilled search space with fewer parameters. Alongside feature interaction selection, some works like AutoFeature [9] and OptInter [15] aim to search for the interaction operation. However, all these works focus on evaluating the whole feature field as a whole instead of individual feature value. To be more precise, the entire *user id* or *City* feature field would be retained or discarded as a unit instead of considering individual feature values like *User x479bs* or *New Orleans*. Considering only feature fields inevitably be coarse-grained, which may overlook informative values in uninformative fields, and vice versa.

In this work, we propose extending the selection granularity of feature interactions from the field to the value level. The extension of granularity would significantly increase the complexity of the entire selection space, leading to an increase in exploration time and memory usage. We manage this challenge by decomposing the selection space using tensor factorization and calculating the corresponding parameters on the fly. To further improve the selection efficiency, we introduce a hybrid-grained feature interaction selection space, which explicitly considers the relation between field-level and value-level. To perform feature interaction selection, we develop a sparsification-based selection algorithm named **OptFeature**(short for **Opt**imizing **Feature** Interaction Selection), which efficiently selects the feature interaction concurrently from both feature fields and feature values. We conduct experiments over three large-scale real-world benchmarks and compare accuracy and efficiency with

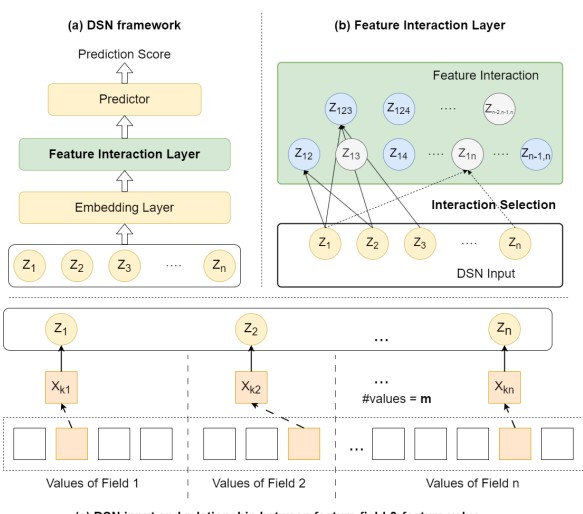

Figure 1: Illustration of deep sparse network and feature interaction layer

state-of-the-art models. Empirical results demonstrate the superiority of our method on both dimensions. Additional studies support the feasibility of our method.

## 2 Related Work

### 2.1 Deep Sparse Networks (DSNs)

It is commonly believed that feature interaction is the critical challenge towards accurate prediction in deep sparse networks [15]. Various models have been proposed to address the feature interaction layer. DeepFM [8] utilizes a factorization machine [21] to model all second-order feature interactions, followed by a multi-layer perceptron as the predictor. More sophisticated feature interaction layers have been proposed to replace the factorization machine, such as the inner product operation in IPNN [18], the cross-network in DCN [24, 25] or the MLP component in PIN [19].

With the advancement of neural architecture search [31, 14, 12] and continuous sparsification [1], various methods have been proposed to select the informative feature interactions [11, 7] and reduce computational costs. AutoFis [11] employs a sparse optimizer to select suitable feature interactions at the field level. PROFIT [7] formulates field-level feature interaction selection within a distilled search space. It employs a progressive search for efficient exploration. AutoIAS [26] takes one step

further to integrate the feature interaction selection as part of the search space and jointly conduct the search with other components like MLP architecture or embedding dimension. GAIN [13], on the other hand, focuses on the DCN [24]-like architectures and conducts the feature interaction selection jointly with the model training. However, all previous works conduct feature interaction selection on the field level. Our work builds on the existing approaches for modelling feature interactions in DSNs. We extend the selection granularity to the value level and propose a hybrid-grained selection approach.

Furthermore, there also exists some works such as OptInter [15], AutoFeature [9], and NAS-CTR [29] take a different angle and search for suitable operations (such as inner product, outer product or element-wise sum) to model each feature interaction properly. These works are perpendicular to our study.

## 2.2 Feature Interaction Selection in DSNs

To conduct feature interaction selection in DSNs, previous works [11, 7, 26, 13] borrow the ideas from neural architecture search [26, 11] and continuous sparsification [7, 13]. We briefly introduce these two techniques in the following. Previous works focusing on feature selection [3] are excluded from the discussion.

### 2.2.1 Neural Architecture Search

Neural architecture search (NAS) automatically identifies optimal architectures for specific tasks and datasets, eliminating the need for expert design [31, 12, 14, 20]. This approach has led to substantial advancements in various domains, including vision [31, 12, 14], natural language processing [23] and sparse representation learning [11, 7]. There are two key aspects to neural architecture search: *search space* and *search algorithm*. The *search space* comprises all potential architectures and is usually task-dependent. An effective search space should be neither too complex, leading to high search costs, nor too shallow, ensuring the inclusion of suitable architectures. The *search algorithm* explores the search space to find appropriate architectures. Search algorithms can generally be categorized into controller-based [31], evaluation-based [20], and gradient-based [14, 12] classes. Our work distinguishes itself from existing research by addressing the challenge of feature interaction selection for deep sparse networks, where the primary obstacles involve high search costs and expansive search spaces.

### 2.2.2 Continuous Sparsification

Continuous sparsification focuses on reducing a continuous value to a sparse form and is closely related to our method [1]. It is often used to prune less informative components or parameters from neural networks, which is intrinsically equivalent to solving an $L_0$ normalization problem. This technique has been leveraged in various aspects of deep sparse networks (DSNs), notably in the embedding layer [17] and the feature interaction layer [11]. Our work employs continuous sparsification to discretize the selection of value-level feature interactions during the search process. By doing so, we can effectively manage the complexity of the search space and maintain a high level of efficiency in the search algorithm.

## 3 Methods

This section will first formulate the general DSNs in Section 3.1. Then it will elaborate on the feature interaction selection problem from field and value perspectives in Section 3.2. We further concrete the proposed OptFeature method, which conducts hybrid-grained feature interaction selection in Section 3.3.

### 3.1 DSNs Formulation

**Feature Field and Value** The input of DSNs comprises multiple feature fields, each containing a set of feature values. Hence, two forms of expression exist for each data sample, deriving from feature field and feature value perspectives. More precisely, a data sample $\mathbf{x}$ consists of $n$ feature fields can be written as:

$$\begin{aligned}
\textit{field perspective:} \quad & \mathbf{x} = [\mathbf{z}_1, \mathbf{z}_2, ..., \mathbf{z}_n], \\
\textit{value perspective:} \quad & \mathbf{x} = [x_{k_1}, x_{k_2}, ..., x_{k_n}].
\end{aligned} \tag{1}$$

The notation $\mathbf{z}_i = \{x_{k_i} \mid k_i \in [1, m_i], k_i \in N_+\}$ represents the set of feature values within the $i$-th field. Here $m_i$ denotes the number of values in each field, and $m = \Sigma_{i=1}^n m_i$ refers to that across all fields. An illustration figure is shown in Figure 1(c). We highlight this formulation as it is one of the core differences between our method, which focuses on both *field* and *value* perspectives, and previous works [11, 7], which focuses only on the *field* perspective. This allows us to capture more granular and informative interactions, potentially leading to better model performance.

**Deep Sparse Networks** As mentioned in Section 1, the DSN commonly consists of three major components: the embedding layer, the feature interaction layer and the predictor. The embedding layer usually employs an embedding table to convert $\mathbf{z}_i$ from high-dimensional, sparse vectors into low-dimensional, dense embeddings. This can be formulated as:

$$\mathbf{e}_i = \mathbf{E} \times \mathbf{z}_i = \mathbf{E} \times x_{k_i}, \tag{2}$$

where $\mathbf{E} \in \mathbb{R}^{m \times d}$ is the embedding table, $d$ is the pre-defined embedding size. These embeddings are further stacked together as the embedding vector $\mathbf{e} = [\mathbf{e}_1, \mathbf{e}_2, ..., \mathbf{e}_n]$, which also serves as the input for feature interaction layer.

The feature interaction layer further performs interaction operations over its input. The $t$-th order feature interaction can be generally represented as:

$$\mathbf{v}^t = \mathcal{V}^t(\mathbf{e}), \tag{3}$$

where $\mathcal{V}(\cdot)$, as the $t$-th order interaction operation, $2 \leq t \leq n$, can vary from a multiple layer perceptron [19] to cross layer [24]. In this paper, we select the inner product, which is recognized to be the most commonly used operation for feature interaction [18, 7], as our interaction operation. For $t$-th order feature interactions, cardinality $|\mathbf{v}^t| = C_n^t$. Note that the selection of $t$ is usually task-dependent. For instance, in the click-through rate prediction, $t = 2$ is considered sufficient [15].

In the prediction layer, both the embedding vector $\mathbf{e}$ and feature interaction vectors $\mathbf{v}^t$ are aggregated together to make the final prediction $\phi_\theta(\mathbf{e}, \{\mathbf{v}^t \mid 2 \leq t \leq n\})$, where $\phi_\theta(\cdot)$ represents the prediction function parameterized on $\theta$. The mainstream prediction function is MLP [18, 19]. Finally, the loss function is calculated over each data sample and forms the training object:

$$\mathcal{L}(\mathcal{D}_{\text{tra}} \mid \mathbf{W}) = \frac{1}{|\mathcal{D}_{\text{tra}}|} \sum_{(\mathbf{x},y) \in \mathcal{D}_{\text{tra}}} \ell(y, \hat{y}) = \frac{1}{|\mathcal{D}_{\text{tra}}|} \sum_{(\mathbf{x},y) \in \mathcal{D}_{\text{tra}}} \ell(y, \phi_\theta(\mathbf{E} \times \mathbf{x}, \{\mathbf{v}^t \mid 2 \leq t \leq n\})), \tag{4}$$

where $\mathbf{W} = \{\mathbf{E}, \theta\}$ denotes the model parameters, $\mathcal{D}_{\text{tra}}$ refers to the training set.

### 3.2 Feature Interaction Selection

Feature interaction selection is crucial in DSNs [11, 7]. In general DSNs, all feature interactions, whether informative or not, are included as inputs for the final predictor. This indiscriminate inclusion of all interactions inevitably introduces noise into the model, which can hinder its performance [11, 7]. Selecting informative feature interactions is called *feature interaction selection*. Proper feature interaction selection can significantly improve the accuracy and efficiency of DSNs by reducing the amount of noise and focusing on learning the most informative interactions [11].

For a $t$-th order feature interaction $\mathbf{v}^t = \{\mathbf{v}_{(i_1,i_2,\cdots,i_t)}\}$, $1 \leq i_t \leq m_{i_t}$, we aim to determine whether to keep each element or not. We take $t = 2$ as an example to illustrate the problem. The selection can be formulated as learning an gate weight $\mathbf{a}_{(i_1,i_2)} \in \{0, 1\}$ corresponding to each $\mathbf{v}_{(i_1,i_2)}$. All $\mathbf{a}_{(i_1,i_2)}$ compose the second-order feature interaction selection tensor $\mathbf{A}^2$, which is formulated as:

$$\mathbf{A}^2 = \begin{bmatrix}
I & \mathbf{a}_{(1,2)} & \mathbf{a}_{(1,3)} & \cdots & \mathbf{a}_{(1,n)} \\
\mathbf{a}_{(1,2)}^T & I & \mathbf{a}_{(2,3)} & \cdots & \mathbf{a}_{(2,n)} \\
\mathbf{a}_{(1,3)}^T & \mathbf{a}_{(2,3)}^T & I & \cdots & \mathbf{a}_{(3,n)} \\
\vdots & \vdots & \vdots & \ddots & \\
\mathbf{a}_{(1,n)}^T & \mathbf{a}_{(2,n)}^T & \mathbf{a}_{(3,n)}^T & \cdots & I
\end{bmatrix}. \tag{5}$$

Here the tensor is symmetric due to the feature interaction being invariant to perturbation and the diagonal is composed of identical tensors. According to different perspectives in Equation 5, there are two forms for Equation 1: (i) when $\mathbf{a}_{(i_1, i_2)}$ is a scalar, $\mathbf{A}^2$ is a 2-$D$ matrix which indicates the field level interaction selection $\mathbf{A}_f^2 \in \{0, 1\}^{n^2}$. (ii) when $\mathbf{a}_{(i_1, i_2)} \in \{0, 1\}^{m_{i_1} \times m_{i_2}}$ is a tensor, $A^2$ is a 2-$D$ matrix which indicates the value level interaction selection $\mathbf{A}_v^2 \in \{0, 1\}^{m^2}$. Therefore, feature interaction selection tensor $\mathbf{A}^t$ of different order $t$ compose the selection tensor set $\mathbf{A} = \{\mathbf{A}^t \mid 2 \leq t \leq n\}$, which is the learning goal of the *feature interaction selection* problem.

**Field-grained**  In previous methods [7, 11], the feature interaction selection is usually conducted at the field level. This can be formulated as learning the field-grained selection tensor set $\mathbf{A} = \{\mathbf{A}_f^t \mid \mathbf{A}_f^t \in \{0, 1\}^{n^t}, 2 \leq t \leq n\}$, where $\mathbf{A}_f^t$ refers to the field-grained feature interaction selection tensor at $t$-th order. Finding a suitable $\mathbf{A}_f^t$ is an NP-hard problem [7]. Some works have been devoted to investigating the search strategy. AutoFis [11] intuitively assigns a continuous tensor $\mathbf{C}_f^t \in \mathbb{R}^{n^t}$ and adopts a sparse optimizer for continuous sparsification so that $\mathbf{C}_f^t$ will approximate $\mathbf{C}_f^t$ eventually. PROFIT [7], on the other hand, uses symmetric CP decomposition [10] to approximate $\mathbf{A}_f^t$ result. This can be formulated as $\hat{\mathbf{A}}_f^t \approx \sum_{r=1}^R \underbrace{\boldsymbol{\beta}^r \circ \cdots \circ \boldsymbol{\beta}^r}_{t \text{ times}}$, where $\circ$ denotes broadcast time operation, $\boldsymbol{\beta}^r \in \mathbb{R}^{1 \times n}$ refers to the decomposed vector and $R \ll n$ is a pre-defined positive integer.

**Value-grained**  To extend the feature interaction selection from field-grained to value-grained, we focus on the value-grained selection tensor set $\mathbf{A} = \{\mathbf{A}_v^t \mid \mathbf{A}_v^t \in \{0, 1\}^{m^t}, 2 \leq t \leq n\}$. Although this formulation is analogous to the field-level one, it is substantially more complex given $m \gg n$. This expansion in complexity poses a challenge, referred to as the *size explosion* problem, as the computational cost and memory usage for the value-level selection increases dramatically. So it is hard to utilize previous methods [11, 7] directly into solving the value-grained selection problem. We will discuss how we tackle this *size explosion* problem in Section 3.3.1 in detail.

**Hybrid-grained**  Apart from learning the field and value-grained selection tensor $\mathbf{A}_f$ and $\mathbf{A}_v$, the hybrid-grained selection also needs to learn a proper hybrid function, we formulate the hybrid function as a binary selection equation,

$$\mathbf{A} = \{\mathbf{A}^t \mid 2 \leq t \leq n\}, \ \mathbf{A}^t = \boldsymbol{\alpha}^t \mathbf{A}_f^t + (1 - \boldsymbol{\alpha}^t) \mathbf{A}_v^t \tag{6}$$

where tensor $\boldsymbol{\alpha}^t \in \{0, 1\}^{n^t}$ denoting the hybrid choice. As the quality of the field and value-grained selection tensors will heavily influence the hybrid tensor $\boldsymbol{\alpha}^t$, joint optimization of both selection tensors $\mathbf{A}_f$ & $\mathbf{A}_v$ and hybrid tensor $\boldsymbol{\alpha}$ is highly desired. We will discuss how we tackle this *joint training* problem in Section 3.3.2 in detail.

Previous field-grained and value-grained selection can be viewed as a special case for Equation 6. By setting the hybrid tensor $\boldsymbol{\alpha}$ in Equation 6 as an all-one tensor $\mathbf{1}$, the original hybrid-grained selection problem is reduced to a field-grained one. Similarly, the value-grained selection problem can also be obtained from Equation 6 by assigning the hybrid tensor $\boldsymbol{\alpha}$ as an all-zero tensor $\mathbf{0}$.

### 3.3 OptFeature

In this section, we formalize our proposed method **OptFeature** aiming to tackle the hybrid-grained feature interaction selection problem depicted in Section 3.2. Such a problem contains two critical issues:

- *Size explosion* problem: As the number of feature values $m$ is significantly larger than the number of feature fields $n$, denoted as $m \gg n$, the value-grained selection tensor $\mathbf{A}_v$ increase dramatically and is hard to be stored or explored. To provide a concrete example regarding the size explosion, we take the commonly used benchmark Criteo[2] as an example. This dataset contains 39 feature fields and $6.8 \times 10^6$ feature values. Even if we only consider $t = 2$ for the second-order feature interactions, the corresponding selection tensor increase from $\|\mathbf{A}_f^2\| = 39^2 = 1521$ to $\|\mathbf{A}_v^2\| =$

---

[2]https://www.kaggle.com/c/criteo-display-ad-challenge

$6.8 \times 10^{6^2} = 4.6 \times 10^{13}$, making it impossible to be stored or explored using vanilla NAS approaches [12, 31, 20] or continuous sparsification methods [1, 6].

- *Joint training* problem: As the quality of the field and value-grained selection tensors $\mathbf{A}_f^t$ & $\mathbf{A}_v^t$ will heavily influence the hybrid tensor $\boldsymbol{\alpha}$, jointly optimization of both selection tensors and hybrid tensor is required. How to efficiently optimize so many large binary tensors remains unsolved.

To address the two critical issues mentioned in Section 3.2 and above, we propose our method **OptFeature** with 1) the *selection tensor decomposition* to address the *size explosion* issue and 2) *sparsification-based selection algorithm* for the *joint training* problem.

### 3.3.1 Selection Tensor Decomposition

To efficiently explore a large space $\mathbf{A}_v$, we first decompose the feature interaction selection tensor. Without the loss of generality, we only focus on the second order, i.e. $t = 2$, value-grained feature interaction selection tensor $\mathbf{A}_v^2$. More details of the general cases $t \geq 2$ are listed in the Appendix A. We omit the superscript for simplicity and use $\mathbf{A}_v$ instead of $\mathbf{A}_v^2$ in later sections. Given that the selection tensor is semi-positive and symmetric, we have the Takagi factorization [2] as:

$$\mathbf{A}_v \approx \mathcal{U} \times \boldsymbol{\Sigma}_v \times \mathcal{U}^T. \tag{7}$$

Here $\boldsymbol{\Sigma}_v$ is a $d' \times d'$ diagonal tensor, $\mathcal{U} \in R^{m \times d'}$ and $d' < m$. To further reduce the consumption of memory and reduce factorization error, we introduce the deep multi-mode tensor factorization [5] that replaces the $\mathcal{U}$ as an output of one neural network, denoted as:

$$\mathcal{U} \approx f_{\theta_v}(\mathbf{E}_v), \tag{8}$$

where $f_{\theta_v} : \mathbb{R}^{m \times \hat{d}} \to \mathbb{R}^{m \times d'}$, $\hat{d} \ll d'$ is a neural network with parameter $\theta_v$ and $\mathbf{E}_v \in \mathbb{R}^{m \times \hat{d}}$ is an additional embedding table for generating feature interaction selection tensor. Figure 2 shows a detailed illustration of our decomposition.

Notably, during the training and inference stage, we do not need to calculate the entire metric $\mathbf{A}$ all in once. We only need to calculate a batch of data in each trial. The element of architecture metric $\mathbf{A}_v[k_i, k_j]$ can be calculated given the following equation:

$$\mathbf{A}_v[k_i, k_j] = f_{\theta_v}(\mathbf{E}_v[k_i, :]) \times \boldsymbol{\Sigma}_v \times f_{\theta_v}^T(\mathbf{E}_v[k_j, :]). \tag{9}$$

The original value-grained selection tensor $\mathbf{A}_v$ consists of $\mathcal{O}(m^2)$ elements. By introducing the Takagi factorization [2], the number of trainable elements is reduced to $\mathcal{O}(md')$. We further reduce that number to $\mathcal{O}(\hat{d}(m + d'))$ through the multi-mode tensor factorization [5]. Hence, we can train the neural network over the value-level selection on modern hardware.

Figure 2: Illustration of the selection tensor decomposition

### 3.3.2 Sparsification-based Selection Algorithm

After decomposing the selection tensor, we still need to jointly conduct feature interaction selection and train the model parameters for an efficient search. To help convert the continuous feature selection tensor into an accurate binary selection, we adopt the straight-through estimator(STE) function [1] for continuous sparsification. The STE can be formulated as a customized function $\mathcal{S}(\cdot)$, with its forward pass as a unit step function

$$\mathcal{S}(x) = \begin{cases} 0, & x \leq 0 \\ 1, & x > 0 \end{cases}, \tag{10}$$

and backward pass as $\frac{d}{dx}\mathcal{S}(x) = 1$, meaning that it will directly pass the gradient backward. Therefore, we can mimic a discrete feature interaction selection while providing valid gradient information for the value-level selection parameters $\mathbf{E}_v$ & $\theta_v$, making the whole process trainable. Hence, Equation 9 is re-written as:

$$\mathbf{A}_v[k_i, k_j] = S(f_{\theta_v}(\mathbf{E}_v[k_i, :]) \times \boldsymbol{\Sigma}_v \times f_{\theta_v}^T(\mathbf{E}_v[k_j, :])). \tag{11}$$

Similar to Equation 11, we can obtain the field-level selection matrix $\mathbf{A}_f$ using the following equation:

$$\mathbf{A}_f[i,j] = S(g_{\theta_f}(\mathbf{E}_f[i,:]) \times \mathbf{\Sigma}_f \times g_{\theta_f}^T(\mathbf{E}_f[j,:])), \tag{12}$$

where $\mathbf{E}_f \in \mathbb{R}^{n \times \hat{d}}$ is the field-level embedding table for feature interaction selection and $g_{\theta_f} : \mathbb{R}^{n \times \hat{d}} \to \mathbb{R}^{n \times d'}$ is another neural network parameterized by $\theta_f$.

After obtaining the value-level selection matrix $\mathbf{A}_v$ and field-level selection matrix $\mathbf{A}_f$, we need to merge them and obtain the hybrid selection result $\mathbf{A}$. Inspired by DARTS [12] and its success in previous works [11, 15, 13], we relax the hybrid tensor $\alpha$ into a continuous tensor $\alpha_c \in \mathbb{R}^{n^t}$, which can be trained jointly with other parameters via gradient descent. To ensure convergence, we apply the sigmoid function $\sigma(\cdot)$ over $\alpha_c$. Hence, during training time, Equation 6 can be re-written as:

$$\mathbf{A} = \sigma(\alpha_c) \cdot \mathbf{A}_f + (1 - \sigma(\alpha_c)) \cdot \mathbf{A}_v. \tag{13}$$

Hence, the final search objective can be formulated as:

$$\hat{\mathbf{W}}^* = \min_{\hat{\mathbf{W}}} \mathcal{L}(\mathcal{D}_{\text{tra}}|\{\mathbf{W}^*, \hat{\mathbf{W}}\}), \text{ s.t. } \mathbf{W}^* = \min_{\mathbf{W}} \mathcal{L}(\mathcal{D}_{\text{val}}|\{\mathbf{W}, \hat{\mathbf{W}}^*\}). \tag{14}$$

Here $\hat{\mathbf{W}} = \{\mathbf{E}_v, \theta_v, \mathbf{E}_f, \theta_f, \alpha\}$ denoting the parameter required in the interaction selection process, and $\mathcal{D}_{\text{val}}$ denotes the validation set. We summarize the overall process of our *sparsification-based selection algorithm* in Algorithm 1.

---

**Algorithm 1** Sparsification-based Selection Algorithm

---

**Require:** training and validation set $\mathcal{D}_{\text{tra}}$ and $\mathcal{D}_{\text{val}}$
**Ensure:** main model parameters $\mathbf{W}$ and feature interaction selection parameters $\hat{\mathbf{W}}$
 1: **for** epoch = 1, $\cdots$, T **do**
 2:     **while** epoch not finished **do**
 3:         Sample mini-batch $\mathcal{B}_{\text{val}}$ and $\mathcal{B}_{\text{tra}}$ from validation set $\mathcal{D}_{\text{tra}}$ and training set $\mathcal{D}_{\text{val}}$
 4:         Update selection parameter $\hat{\mathbf{W}}$ using gradients $\nabla_{\hat{\mathbf{W}}} \mathcal{L}(\mathcal{B}_{\text{val}}|\{\mathbf{W}, \hat{\mathbf{W}}\})$
 5:         Update model parameter $\mathbf{W}$ using gradients $\nabla_{\mathbf{W}} \mathcal{L}(\mathcal{B}_{\text{tra}}|\{\mathbf{W}, \hat{\mathbf{W}}\})$
 6:     **end while**
 7: **end for**

---

### 3.3.3 Retraining with Optimal Selection Result

During the re-training stage, the optimal hybrid tensor is determined as $\alpha^* = \mathbb{1}_{\alpha_c > 0}$ following previous works [12, 15, 13]. Hence, we can obtain the optimal selection result

$$\mathbf{A}^* \approx \alpha^* \cdot S(g_{\theta_f^*}(\mathbf{E}_f^*) \times \mathbf{\Sigma}_f^* \times g_{\theta_f^*}^T(\mathbf{E}_f^*)) + (1 - \alpha^*) \cdot S(f_{\theta_v^*}(\mathbf{E}_v^*) \times \mathbf{\Sigma}_v^* \times f_{\theta_v^*}^T(\mathbf{E}_v^*)). \tag{15}$$

We freeze the selection parameters $\hat{\mathbf{W}}$ and retrain the model from scratch following the existing works [12, 7]. This reason for introducing the retraining stage is to remove the influence of sub-optimal selection results during the selection process over the model parameter. Also,

## 4 Experiment

### 4.1 Experiment Setup

**Datasets** To validate the effectiveness of our proposed method OptFeature, we conduct experiments on three benchmark datasets (Criteo, Avazu[3], and KDD12[4]), which are widely used in previous work on deep sparse networks for evaluation purposes [7, 11]. The dataset statistics and preprocessing details are described in the Appendix B.

---

[3]http://www.kaggle.com/c/avazu-ctr-prediction
[4]http://www.kddcup2012.org/c/kddcup2012-track2/data

**Baseline Models**   We compare our OptFeature with previous works in deep sparse networks. We choose the methods that are widely used and open-sourced:

- Shallow network: Logistic Regression(LR) [22] and Factorization Machine(FM) [21].

- DSNs: FNN [27], DeepFM [8], DCN [24], IPNN [18].

- DSNs with feature interaction selection(DSNs with FIS): AutoFIS[5] [11] and PROFIT[6] [7].

For the shallow networks and deep sparse networks, we implement them following the commonly used library torchfm [7]. These baselines and our OptFeature are available here[8]. For AutoFIS and PROFIT, we re-use their original implementations, respectively.

**Evaluation Metrics**   Following the previous works[8, 30], we adopt the commonly-used performance evaluation metrics for click-through rate prediction, **AUC** (Area Under ROC curve) and **Log loss** (also known as cross-entropy). As for efficiency evaluation, we also use two standard metrics, inference time and model size, which are widely adopted in previous works of deep sparse network [7, 11].

**Parameter Setup**   To ensure the reproducibility of experimental results, here we further introduce the implementation setting in detail. We implement our methods using PyTorch. We adopt the Adam optimizer with a mini-batch size of 4096. We set the embedding sizes to 16 in all the models. We set the predictor as an MLP model with [1024, 512, 256] for all methods. All the hyper-parameters are tuned on the validation set with a learning rate from [1e-3, 3e-4, 1e-4, 3e-5, 1e-5] and weight decay from [1e-4, 3e-5, 1e-5, 3e-6, 1e-6]. We also tune the learning ratio for the feature interaction selection parameters from [1e-4, 3e-5, 1e-5, 3e-6, 1e-6] and while weight decay from [1e-4, 3e-5, 1e-5, 3e-6, 1e-6, 0]. The initialization parameters for the retraining stage are selected from the best-performed model parameters and randomly initialized ones.

**Model Stability**   To make the results more reliable, we ran the repetitive experiments with different random seeds five times and reported the average value for each result.

**Hardware Platform**   All experiments are conducted on a Linux server with one Nvidia-Tesla V100-PCIe-32GB GPU, 128GB main memory and 8 Intel(R) Xeon(R) Gold 6140 CPU cores.

## 4.2   Benchmark Comparison

**Model Performance**   We present the model performance across three benchmark datasets in Table 1 and make the following observations. Firstly, OptFeature consistently outperforms other models on all three benchmark datasets, validating the effectiveness of our hybrid-grained selection space and selection algorithm. Secondly, DSNs generally yield better performances than shallow models, underscoring the importance of designing powerful sparse networks. Thirdly, DSNs with feature interaction selection tend to perform superiorly compared to those incorporating all possible interactions. This confirms the significance of conducting feature interaction selection within DSNs. Lastly, methods that decompose the selection space (e.g., OptFeature and PROFIT) consistently outperform AutoFIS, which directly navigates the original selection space.

**Model Efficiency**   We also evaluate model efficiency and present the results in Figure 3. We can observe that OptFeature outperforms all other methods in terms of inference time, while its model size is comparable to other feature interaction selection methods. Both model accuracy and inference efficiency are critical factors when deploying DSNs. Our OptFeature not only achieves the best performance but also reduces inference time, indicating its practical value.

---

[5]https://github.com/zhuchenxv/AutoFIS

[6]https://github.com/NeurIPS-AutoDSN/NeurIPS-AutoDSN

[7]https://github.com/rixwew/pytorch-fm

[8]https://github.com/fuyuanlyu/OptFeature

Table 1: Overall Performance Comparison

| Dataset | | Criteo | | Avazu | | KDD12 | |
|---|---|---|---|---|---|---|---|
| Category | Model | AUC | Logloss | AUC | Logloss | AUC | Logloss |
| Shallow | LR | 0.7882 | 0.4609 | 0.7563 | 0.3928 | 0.7411 | 0.1637 |
| | FM | 0.8047 | 0.4464 | 0.7839 | 0.3783 | 0.7786 | 0.1566 |
| DSNs | FNN | 0.8101 | 0.4414 | 0.7891 | 0.3762 | 0.7947 | 0.1536 |
| | DeepFM | 0.8097 | 0.4418 | 0.7896 | 0.3757 | 0.7969 | 0.1531 |
| | DCN | 0.8096 | 0.4419 | 0.7887 | 0.3767 | 0.7955 | 0.1534 |
| | IPNN | 0.8103 | 0.4413 | 0.7896 | 0.3741 | 0.7945 | 0.1537 |
| DSNs with FIS | AutoFIS | 0.8089 | 0.4428 | 0.7903 | 0.3749 | 0.7959 | 0.1533 |
| | PROFIT | 0.8112 | 0.4406 | 0.7906 | 0.3756 | 0.7964 | 0.1533 |
| | OptFeature | **0.8116** | **0.4402** | **0.7925**$^*$ | **0.3741**$^*$ | **0.7982**$^*$ | **0.1529**$^*$ |

Here $^*$ denotes statistically significant improvement (measured by a two-sided t-test with p-value $< 0.05$) over the best baseline. The best and second best performed results are marked in **bold** and underline format

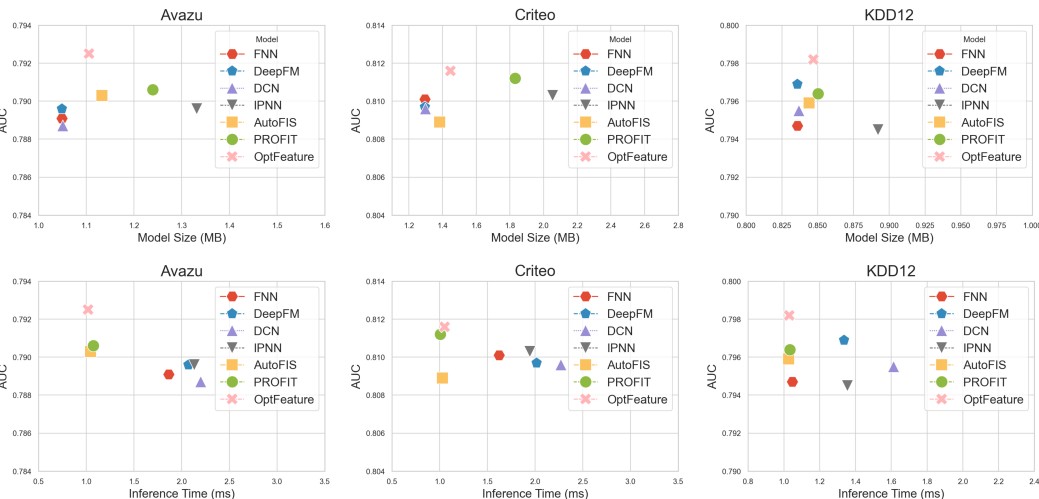

Figure 3: Efficiency comparison between OptFeature and DSN baselines. Model size excludes the embedding layer. Inference time refers to the average inference time per batch across validation set.

## 4.3 Investigating the Selection Process

In this section, we delve into the selection process of OptFeature. We introduce two variants of OptFeature for this purpose: (i) OptFeature-f, which carries out field-level interaction selection, and (ii) OptFeature-v, which conducts value-level interaction selection. Both methods adopt the same selection algorithm outlined in Section 3.3.2. We will compare OptFeature and its two variants to other DSNs interaction selection baselines regarding effectiveness and efficiency.

Table 2: Performance Comparison over Different Granularity.

| Dataset | | Criteo | | Avazu | | KDD12 | |
|---|---|---|---|---|---|---|---|
| Model | Granularity | AUC | Logloss | AUC | Logloss | AUC | Logloss |
| AutoFIS | Field | 0.8089 | 0.4428 | 0.7903 | 0.3749 | 0.7959 | 0.1533 |
| PROFIT | Field | 0.8112 | 0.4406 | 0.7906 | 0.3756 | 0.7964 | 0.1533 |
| OptFeature-f | Field | 0.8115 | 0.4404 | 0.7920 | 0.3744 | 0.7978 | 0.1530 |
| OptFeature-v | Value | 0.8116 | 0.4403 | 0.7920 | 0.3742 | 0.7981 | 0.1529 |
| OptFeature | Hybrid | 0.8116 | 0.4402 | 0.7925 | 0.3741 | 0.7982 | 0.1529 |

**Investigating the Selection Effectiveness** We evaluate the effectiveness of various DSN feature interaction methods in terms of comparable performance. Several observations can be made from the results presented in Table 2. Firstly, OptFeature consistently delivers superior results on all three datasets. This confirms that the hybrid-grained selection space allows OptFeature to explore finer-grained while the selection algorithm effectively performs based on the decomposition. Secondly, the value-level selection method consistently surpasses field-level selection, suggesting that field-level selection might be too coarse-grained to leverage informative values within uninformative fields. Lastly, OptFeature-f outperforms both AutoFIS and PROFIT in terms of model performance.

The distinguishing factor among these three methods lies in the selection algorithm. AutoFIS directly optimizes the gating vector for each interaction field, rendering the space too large for efficient exploration. PROFIT, on the other hand, adopts a progressive selection algorithm, leading to sub-optimal interaction selection. This finding further emphasizes the superiority of our sparsification-based selection algorithm in selecting informative interactions.

**Investigating the Selection Efficiency**  Next, we perform a comparison study on the efficiency of the selection methods. In Figure 4, we report the search cost, measured in GPU hours. The search cost reflects the GPU resource consumption when selecting interactions from scratch. It's important to note that the cost of re-training is excluded as our focus is on the selection process. We observe that the search cost of OptFeature is lower than other baselines on the Criteo and KDD12 datasets. OptFeature surpasses the other two variants as it typically converges in fewer epochs. These comparisons underscore the feasibility of our proposed method. However, on the Avazu dataset, OptFeature is slightly outperformed by its two variants, i.e., OptFeature-f and OptFeature-v, because this dataset usually converges in one epoch [28]. As a result,

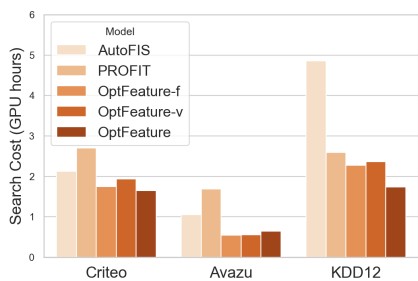

Figure 4: Comparisons between OptFeature and other DSNs interaction selection baselines on the search cost.

all three methods converge within the same epoch. Furthermore, compared to AutoFIS and PROFIT, OptFeature-f incurs a lower cost with the same complexity of selection space. This observation further highlights the superiority of our selection algorithm in terms of convergence speed.

## 5  Conclusions and Limitations

In this work, we introduce a hybrid-grained selection approach targeting both feature field and feature value level. To explore the vast search space that extends from field to value, we propose a decomposition method to reduce space complexity and make it trainable. We then developed a selection algorithm named OptFeature that simultaneously selects interactions from field and value perspectives. Experiments conducted on three real-world benchmark datasets validate the effectiveness and efficiency of our method. Further ablation studies regarding search efficiency and selection granularity illustrate the superiority of our proposed OptFeature.

**Limitations**  Despite OptFeature demonstrating superior performance over other baselines on model performance and inference time, it requires a larger model size than certain DSNs [8, 27]. Additionally, it lacks online experiments to validate its effectiveness in more complex and dynamic scenarios.

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

# A  General Form of Selection Tensor Decomposition

In this section, we further extend the selection tensor decomposition in Section 3.3.1 from a special case, where $t = 2$, to a more general case, where $2 \leq t \leq n$. The interaction selection tensor $\mathbf{A}_v^t$ for $t$-th order features is also semi-positive and symmetric. By extending the Takagi factorization [2], we have:

$$\mathbf{A}_v^t = \mathbf{\Sigma}_v \times_1 \mathcal{U} \times_2 \mathcal{U} \times_3 \cdots \times_t \mathcal{U}, \tag{16}$$

where $\mathbf{\Sigma}_v$ is a $\underbrace{d' \times \cdots \times d'}_{t \text{ times}}$ diagonal tensor, $\times_t$ denotes the $t$-mode matrix multiplication [5], $\mathcal{U} \in R^{m \times d'}$ and $d' < m$. Similar to the special case where $t = 2$, we adopt multi-mode tensor factorization [5] to replace $\mathcal{U}$ as an output of a neural network, denoted as:

$$\mathcal{U} \approx f_{\hat{\theta}}(\mathbf{E}_v), \tag{17}$$

where $f_{\hat{\theta}} : \mathbb{R}^{m \times \hat{d}} \rightarrow \mathbb{R}^{m \times d'}$, $\hat{d} \ll d'$ is a neural network with parameter $\hat{\theta}$ and $\hat{\mathbf{E}} \in \mathbb{R}^{m \times \hat{d}}$ is an additional embedding table for generating feature interaction selection tensor. The element of architecture metric $\mathbf{A}_v^t[k_{i_1}, \cdots, k_{i_t}]$ can be calculated given the following equation:

$$\mathbf{A}_v^t[k_{i_1}, \cdots, k_{i_t}] = \mathbf{\Sigma}_v \times_1 f_{\theta_v}(\mathbf{E}_v[k_{i_1}, :]) \times_2 \cdots \times_t f_{\theta_v}(\mathbf{E}_v[k_{i_t}, :]). \tag{18}$$

The original value-grained selection tensor $\mathbf{A}_v^t$ consists of $\mathcal{O}(m^t)$ elements. The trainable elements is reduced to $\mathcal{O}(md')$ after the Takagi factorization [2] and to $\mathcal{O}(\hat{d}(m + d'))$ after the multi-mode tensor factorization [5].

# B  Dataset and Preprocessing

We conduct our experiments on two public real-world benchmark datasets. The statistics of all datasets are given in Table 3. We describe all these datasets and the pre-processing steps below.

Table 3: Dataset Statistics

| Dataset | #samples | #field | #value | pos ratio |
|---|---|---|---|---|
| Criteo | $4.6 \times 10^7$ | 39 | $6.8 \times 10^6$ | 0.2562 |
| Avazu | $4.0 \times 10^7$ | 24 | $4.4 \times 10^6$ | 0.1698 |
| KDD12 | $1.5 \times 10^8$ | 11 | $6.0 \times 10^6$ | 0.0445 |
| Industrial | $3.0 \times 10^8$ | 134 | 2498 | 0.1220 |

Note: *#samples* refers to the total samples in the dataset, *#field* refers to the number of feature fields for original features, *#value* refers to the number of feature values for original features, *pos ratio* refers to the positive ratio.

**Criteo** dataset consists of ad click data over a week. It consists of 26 categorical feature fields and 13 numerical feature fields. Following the best practice [30], we discretize each numeric value $x$ to $\lfloor \log^2(x) \rfloor$, if $x > 2$; $x = 1$ otherwise. We replace infrequent categorical features with a default "OOV" (i.e. out-of-vocabulary) token, with *min_count*=2.

**Avazu** dataset contains 10 days of click logs. It has 24 fields with categorical features. Following the best practice [30], we remove the *instance_id* field and transform the *timestamp* field into three new fields: *hour*, *weekday* and *is_weekend*. We replace infrequent categorical features with the "OOV" token, with *min_count*=2.

**KDD12** dataset contains training instances derived from search session logs. It has 11 categorical fields, and the click field is the number of times the user clicks the ad. We replace infrequent features with an "OOV" token, with *min_count*=10.

**Industrial** dataset is a private large-scale commercial dataset. This dataset contains nearly 3.5 million samples with 134 feature fields and 2498 feature values. Please notice that this dataset has more feature fields and fewer feature values, which differs from the previous benchmarks with fewer feature fields and larger feature values. The results of this dataset are shown in Appendix C.5.

# C  Ablation Study

## C.1  Feature Interaction Operation

In this section, we conduct an ablation study on the feature interaction operation, comparing the performance of the default setting, which uses the *inner product*, with the *outer product* operation. We evaluate these operations on *OptFeature* and its two variants: *OptFeature-f* and *OptFeature-v*. The results are summarized in Table 4.

Table 4: Performance Comparison over Different Feature Interaction Operation.

| Dataset | | Criteo | | Avazu | | KDD12 | |
|---|---|---|---|---|---|---|---|
| Category | Model | AUC | Logloss | AUC | Logloss | AUC | Logloss |
| | OptFeature-f | 0.8115 | 0.4404 | 0.7920 | 0.3744 | 0.7978 | 0.1530 |
| inner product | OptFeature-v | 0.8116 | 0.4403 | 0.7920 | 0.3742 | 0.7981 | 0.1529 |
| | OptFeature | 0.8116 | 0.4402 | 0.7925 | 0.3741 | 0.7982 | 0.1529 |
| | OptFeature-f | 0.8114 | 0.4404 | 0.7896 | 0.3760 | 0.7957 | 0.1535 |
| outer product | OptFeature-v | 0.8113 | 0.4405 | 0.7902 | 0.3752 | 0.7961 | 0.1533 |
| | OptFeature | 0.8115 | 0.4403 | 0.7899 | 0.3753 | 0.7961 | 0.1533 |

From the table, we observe that the *inner product* operation outperforms the *outer product* operation. This performance gap is particularly significant on the Avazu and KDD12 datasets, while it is relatively insignificant on the Criteo dataset. The drop in performance with the *outer product* operation is likely due to the introduction of a significantly larger number of inputs into the final predictor. This makes it more challenging for the predictor to effectively balance the information from raw inputs and feature interactions.

## C.2  Dimension Selection

In this section, we perform an ablation study on the feature interaction selection dimension $\hat{d}$. We compare the AUC performance with the corresponding dimension $\hat{d}$ and present the results in Figure 5. From the figure, we can observe that as the dimension $\hat{d}$ increases, the AUC performance remains relatively consistent over the Criteo dataset. This suggests that it is relatively easy to distinguish value-level selection on the Criteo dataset. However, on the Avazu and KDD12 datasets, the AUC performance improves as the selection dimension $\hat{d}$ increases. This indicates that distinguishing informative values is comparatively more challenging on these two datasets.

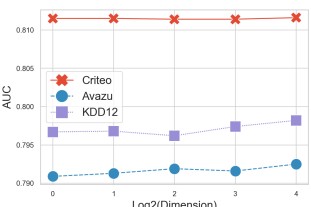

Figure 5: Ablation over feature interaction selection dimension on OptFeature.

## C.3  Higher-Order Feature Interactions

In this section, we investigate the influence of higher-order feature interactions over the final results on the KDD12 dataset. We compare the default setting which only considers second-order interactions with two other settings: (i) only third-order interactions and (ii) both second and third-order interactions. We visualize the result in Figure 6.

From the figure, we can draw the following observations. First, only considering third-order interactions leads to the worst performance. This aligns with the common understanding that second-order interactions are typically considered the most informative in deep sparse prediction [15]. Second, for field-level selection, the performance improves when both second and third-order interactions are incorporated into the model. This finding is consistent with previous studies [11, 7], as the inclusion of additional interactions introduces extra information that enhances the performance. In contrast, for value-level selection, the performance tends to decrease when both second and third-order interactions are included. This could be attributed to the fact that value-level selection operates at a finer-grained level and might be more challenging to optimize directly. Finally, OptFeature constantly outperforms its two variants over all settings. This indicates the feasibility and effectiveness of hybrid-grained selection, which combines both field-level and value-level interactions.

## C.4 Selection Visualization

**Criteo-OptFeature-f**     **Criteo-OptFeature-v**     **Criteo-OptFeature**

(a) Criteo Dataset

**Avazu-OptFeature-f**     **Avazu-OptFeature-v**     **Avazu-OptFeature**

(b) Avazu Dataset

**KDD12-OptFeature-f**     **KDD12-OptFeature-v**     **KDD12-OptFeature**

(c) KDD12 Dataset

Figure 7: Visualization of the Feature Interaction Selection Results.

In this section, we present the visualization of the interaction selection results for OptFeature and its two variants in Figure 7. OptFeature-f performs a binary selection for each interaction field, allowing for easy visualization through a heatmap representation where one indicates keep and zero indicates drop. On the other hand, OptFeature-v and OptFeature involve value-level interaction selection. Hence, we visualize them by setting each element as the percentage of being selected over the training set. The detailed equation for calculating the value for the interaction field (i, j) is shown in Equation 19.

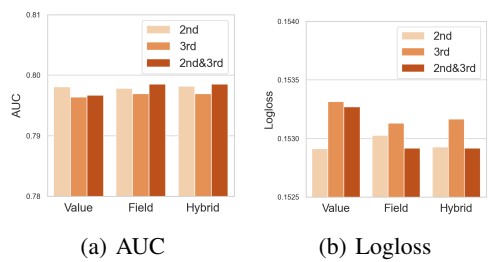

(a) AUC           (b) Logloss

Figure 6: Performance Comparison over Different Feature Interaction Orders.

$$\mathbf{P}_{(i,j)} = \frac{\#\text{Samples keeping interaction field (i, j)}}{\#\text{Training Samples}} \tag{19}$$

From the visualization, we can observe that OptFeature acts as a hybrid approach, exhibiting a combination of both field-level and value-level interactions. Interestingly, we note significant differences between certain interaction fields in the KDD12 and Avazu datasets. OptFeature-f retains all of its interactions, while OptFeature-v only keeps a proportion of the value-level interactions. This observation further emphasizes the importance of exploring interactions at a finer-grained level.

## C.5 Experiment on Industrial Dataset

We explicitly conduct experiments on an industrial large-scale dataset, which contains more feature fields and fewer feature values compared with previous benchmarks. The following results in Table 5 can further prove the effectiveness of OptFeature. We can make the following observations. Firstly, OptFeature outperforms other models on the industrial dataset, validating the effectiveness of our hybrid-grained selection space and selection algorithm. This observation is consistent with those on the other three public benchmarks, as shown in Table 1. Secondly, AutoFIS performs superiorly compared to other baselines. This confirms the significance of conducting feature interaction selection within DSNs. Lastly, PROFIT is surpassed by AutoFIS, unlike on previous public datasets. We believe this is likely due to the increasing number of feature fields. AutoFIS, which treats each field independently, can extend easily to datasets with more feature fields. AutoFIS, which decomposes the feature selection result on the field level, does not generalize well. With the increase of the feature fields, the complexity of such decomposition increases exponentially. OptFeature, on the contrary, can potentially extend to datasets containing more fields better due to its hybrid selection granularity.

Table 5: Performance Comparison on Industrial Dataset

| Dataset | | Industrial | |
| --- | --- | --- | --- |
| Category | Model | AUC | Logloss |
| Shallow | LR | 0.7745 | 0.2189 |
| | FM | 0.7780 | 0.2181 |
| DSNs | FNN | 0.7838 | 0.2168 |
| | DeepFM | 0.7824 | 0.2179 |
| | DCN | 0.7844 | 0.2167 |
| | IPNN | 0.7883 | 0.2147 |
| DSNs with FIS | AutoFIS | 0.7889 | 0.2146 |
| | PROFIT | 0.7849 | 0.2161 |
| | OptFeature | **0.7893** | **0.2142** |

The best and second best performed results are marked in **bold** and underline format respectively.

# D Broader Impact

Successfully identifying informative feature interactions could be a double-edged sword. On the one hand, by proving that introducing noisy features into the model could harm the performance, feature interaction selection could be used as supporting evidence in preventing certain businesses, such as advertisement recommendations, from over-collecting users' information, thereby protecting user privacy. On the other hand, these tools, if acquired by malicious people, can be used for filtering out potential victims, such as individuals susceptible to email fraud. As researchers, it is crucial for us to remain vigilant and ensure that our work is directed towards noble causes and societal benefits.

