# OpenReview forum: "Towards Hybrid-grained Feature Interaction Selection for Deep Sparse Network"
_NeurIPS.cc/2023/Conference — NeurIPS 2023 poster_

### Official Review · Reviewer_eecX · 2023-06-16

**Soundness:** 3 good
**Presentation:** 3 good
**Contribution:** 2 fair
**Rating:** 5
**Confidence:** 4

**Summary:**

The authors propose a novel feature selection algorithm, aiming to detect interactions between features at instance-level, contrary to the usual feature selection algorithms that selects the same features for every sample. Initially, the authors propose a highly memory-demanding approach, requiring an $m \times m$ matrix, being m the number of different values the features can take. Later, a less memory-demanding approach is presented, using matrix decomposition. The experimental results are slightly better to the state-of-the-art.

**Strengths:**

- **Quality:** Several DSN methods were included in the state-of-the-art section. The proposed algorithm is able to obtain very similar results.
- **Clarity:** The paper is easy to follow and to understand. The decisions made are clearly motivated.
- **Significance:** The idea of selecting, per each sample, the most important interactions between features is very interesting and it can provide a good explanation about the decision making.

**Weaknesses:**

- **Originality:** The algorithm is a combination of well-known techniques. The innovative part is focused on how to merge all of them.
- **Quality:** There exists other field of methods that also address feature interaction: the so-called *dynamic feature selection'. Techniques like L2X [1] are focused on the same goal, without the need of using DSNs, which highly reduces the memory consumption. Some information regarding these techniques should be included in the paper.
- **Clarity:** Fig. 3 is clearly misleading. Although it constantly suggest the proposed method outperforms the state-of-the-art, the granularity of the y-axis is almost non-existent. There are very little differences amongst all algorithms.
- **Significance:** I have concerns regarding two critical aspects of the experimental results:
    1. The experimental results show very little improvements against the baseline methods. An statistical analysis is mandatory, in order to establish whether the obtained results provide a real improvement over the state-of-the-art or not.
    2. Although I agree with the authors that feature interaction selection can provide insightful information regarding the decisions provided by the network, the authors do not mention anything related to this in the experimental section.

[1] Chen, J., Song, L., Wainwright, M., & Jordan, M. (2018, July). Learning to explain: An information-theoretic perspective on model interpretation. In International Conference on Machine Learning (pp. 883-892). PMLR.

**Questions:**

See the weaknesses sec

**Limitations:**

Not applied.

---

> ### Author Rebuttal · Authors · 2023-08-10
>
> Hi Reviewer eecX:
>
> Thanks for reviewing our paper and offering helpful comments. Below are responses to your questions.
>
> ### **W1: Originality**
> Please kindly allow me to highlight our papers' originality here, as our writings may not be optimal and can confuse the reviewer. Our initial intuition and major contribution is to extend the granularity of feature interaction selections from field level to value level. To the best of our knowledge, we are the first to propose such an extension. To tackle the *size explosion* problem caused by such an extension, we factorize the selection space. For better *selection efficiency*, we proposed a hybrid-grained selection algorithm, which efficiently selects feature interaction concurrently from both feature field and feature value.
> Hopefully, our response can address the reviewer's concern about our originality, which might be caused by our writing.
>
> ### **W2 and Q1: comparison with techniques from other domains**
> We explicitly want to thank the reviewer for mentioning L2X[3]. L2X is a classic yet inspiring work that utilizes mutual information scores as guidance to conduct *feature selection*. We are informed of this work before our project. However, we decide to exclude the discussion of these works mainly because L2X targets the *feature selection* problem(FS for short), while OptFeature targets the *feature interaction selection* problem(FIS for short).
>
> The difference between FS and FIS generally lies in three aspects. First, the selection space is different. Given $m$ feature values, FS's selection space is smaller than $m$, while FIS's selection space is around $m^t$, with $t$ denoting the order ($t \geq 2$). Second, FIS generally requires fixed features. Third, FIS is considered an important problem in the DSN community, given that feature interaction is generally believed to be an important factor for performance boosting[1,2]. Comparably, FS is a general machine-learning problem.
>
> Although the differences between FS and FIS do not mean we can not borrow intuition from works like L2X, directly borrowing off-the-shell methods might not be practical. Take L2X as an example. Even if we conduct a $t$-th order($t \ge 2)$ field-level FIS using L2X(which is much smaller than value-level FIS), the computation cost for the cross-entropy can be as large as $O(n^t * S)$, with $S$ and $n$ denoting the number of data samples and the number of feature fields in the dataset. Considering the benchmark we adopted contains $n \ge 10$ feature fields and $S \ge 10^7$ data samples (Please see Appendix B1 for details), it can be unbearable for real-world systems. This suggests that we need further investigation into this issue.
>
> ### **W3: Clarity in Fig. 3**
> We thank the reviewer for pointing out the granularity issue of the y-axis in Fig. 3. However, as we can observe, the y-axis in Fig. 3 represents the AUC, which reflects the performance of models. So we think this point refers to the same issue as the following point, which is the significance of our result. Kindly check CW1 in the common response for a more detailed response.
>
> ### **W4 Part 1 and Q2: significance analysis**
> We deeply appreciate the reviewer's effort in keeping this rigid standard of significance analysis, as they are extremely important for the community. Kindly check CW1 in the common response for details. In short, we add a two-tailed t-test to statistically analyze the significance of our result.
>
> ### **W4 Part 2 and Q3: insightful information**
> We try to provide some insightful information in Appendix C4 and Figure 7 due to the page limit. Our message we can easily observe from Figure 7 is that: even though the field-level FIS methods retain certain feature interactions, the value-level FIS method still drops a significant proportion on the value level, resulting in better performance. Such an observation is consistent on all three benchmarks and serves as a justification of our initial intuition: field-level FIS is not good enough.
>
> ### **Q4: benefits with other algorithms**
> We think the major benefit lies in both effectiveness and efficiency. As mentioned, we are the first to bring feature interaction selection to finer-grained and propose a hybrid-selection approach. Such a design can help us to select informative interactions, which leads to better performance. As a side-effect, given the uninformative feature interactions are dropped by our approach, the inference time and model size are slightly improved compared with other SOTA baselines, like PROFIT.
>
> ### Reference:
> 1. Factorization machines (ICDM 2010)
> 2. DeepFM: a factorization-machine based neural network for CTR prediction (IJCAI 2017)
> 3. Learning to explain: An information-theoretic perspective on model interpretation (ICML 2018)
>
> *Please notice that we use the format paper "title(venue)" here due to the page limit. We fully appreciate all author's contributions to the community.*

---

> > ### Comment · Reviewer_eecX · 2023-08-18
> > **Answer to authors**
> >
> > I would like to thank the authors for their responses. Regarding to the Clarity in Fig. 3 issue, the problem is that the differences between the models seems higher than reality. For instance, in the first figure, the y-axis moves between 0.8090 and 0.8115. Visually, a value in the lower bound of the axis should mean a bad performance, and not just a minor decrease in the performance. The margin between the top and the bottom should be higher than the one used in the figures.

---

> > > ### Author Response · Authors · 2023-08-18
> > >
> > > Hi Reviewer eecX,
> > > Sorry for the misunderstanding about your concern on Fig 3. Your argument is valid as the original Fig 3 can be confusing without careful examination. We will change the figure correspondingly as required. Kindly let us know if there is any further concern.

---

> > > > ### Comment · Reviewer_eecX · 2023-08-18
> > > >
> > > > I have raised my score to borderline accept. Although I understand the difference the authors make between feature selection and feature interaction, I still do not see any evidence that suggests the latter provide more informative (or accurate) solutions. Besides that, I think it is an interesting paper.

---

> > > > > ### Author Response · Authors · 2023-08-19
> > > > >
> > > > > Hi Reviewer eecX, thanks for appreciating our paper. We are sorry to hear about your confusion regarding feature selection and feature interaction selection. We will try our best to further clarify these concepts in the later version.

---

> ### Author Response · Authors · 2023-08-18
>
> Hi Reviewer eecX,
> We would again appreciate the valuable and thoughtful review. Since the deadline for the discussion period is approaching, it would be great to have feedback on if our response addresses the concerns raised in your initial review.

---

### Official Review · Reviewer_Bzgv · 2023-07-02

**Soundness:** 2 fair
**Presentation:** 2 fair
**Contribution:** 2 fair
**Rating:** 4
**Confidence:** 4

**Summary:**

This work proposes a hybrid-grained feature interaction selection approach for deep sparse networks, which targets both feature field and feature value. The proposed approach uses a decomposed space that is calculated on the fly to explore the expansive space of feature interactions. The work also introduces a selection algorithm called OptFeature, which efficiently selects the feature interaction from both the feature field and the feature value simultaneously. The proposed approach is evaluated on three large real-world benchmark datasets, and the results demonstrate that the proposed approach performs well in terms of accuracy and efficiency. The work concludes that the proposed approach can effectively select feature interactions in deep sparse networks, and it has the potential to improve the performance of prediction tasks with high-dimensional sparse features.

**Strengths:**

1. The hybrid-grained feature interaction selection approach goes beyond traditional field-level selection, and the decomposed space and sparsification-based selection algorithm make the work appear to be a cutting-edge method to some extent.
2. This work ran the repetitive experiments with different random seeds five times and reported the average value for each result, and provides information about the parameter setup, metrics, datasets, baseline and parameter setup, so the experimental results appear to be reliable.

**Weaknesses:**

1. Novelty: The proposed approach that targets both feature field and feature value levels and introduced a decomposed space and a sparsification-based selection algorithm to explore the selection space, which appears to be a novel contribution to the field. But it does not provide a comprehensive review of related Feature Interaction Selection work in Section 2, and the novelty is not so obvious.

2. Experiments: What GPU was used in this work? How many were used? Were all the experiments conducted on the same GPU? Why formulate the hybrid-grained feature interaction selection as a binary selection according to Equation 6? Taking either 0 or 1 doesn’t seem to reflect ‘hybrid’.

3. Writing: the introduction does not summarize the main contributions of this work, so readers cannot intuitively get the advantages of this work. In addition, the content of Section 2.1 introducing Neural Architecture Search seems to be not very relevant to this paper. Furthermore, Section 3.3.2 does not explain how to determine the parameter α in Equation 6, which makes one wonder how to choose between value-grained and field-grained.




**Questions:**

1. Experiments: What GPU was used in this work? How many were used? Were all the experiments conducted on the same GPU? Why formulate the hybrid-grained feature interaction selection as a binary selection according to Equation 6? Taking either 0 or 1 doesn’t seem to reflect ‘hybrid’.

2. Writing: the introduction does not summarize the main contributions of this work, so readers cannot intuitively get the advantages of this work. In addition, the content of Section 2.1 introducing Neural Architecture Search seems to be not very relevant to this paper. Furthermore, Section 3.3.2 does not explain how to determine the parameter α in Equation 6, which makes one wonder how to choose between value-grained and field-grained.

**Limitations:**

See the weaknesses.

---

> ### Author Rebuttal · Authors · 2023-08-10
>
> Hi Reviewer Bzgv:
>
> Thanks for reviewing our paper and offering detailed comments. Below are responses to your questions.
>
> ### **W1: Unclear expression of our novelty**
> Thanks for recognizing the novelty of our method. This comment does remind us of the importance of constantly highlighting our contribution and clearly differing our method from related works. Please check the common review regarding our revision.
>
> ### **W2 Part 1: Hardware platform**
> We thank the reviewer for highlighting these details, as they can be important for reproducibility and can benefit the whole community. We listed these details below (also detailedly described in Appendix B3). Kindly check if they can address your question. However, the author's comment reminds us to be self-contained in the main paper. We will add corresponding references in the main paper later.
>
> | Hardware | Config |
> | --- | --- |
> | CPU | 8-core Intel(R) Xeon(R) Gold 6140 CPU |
> | GPU | Nvidia-Tesla V100-PCIe-32GB |
> | Memory | 128GB |
> | System | Ubuntu 18.04 LTS |
>
> ### **W2 Part 2: Design of hybrid-grained feature interaction selection**
> Based on our humble opinion, the reviewer's confusion is about what part of the model is "hybrid". In our work, we define "hybrid" as one feature interaction conducts selection on the value level while another conducts selection on the field level. For instance, a 2nd-order feature interaction <*user id*, *City*> may conduct selection on the value level, as the user's preference may differ a lot. Meanwhile, another 2-nd order feature interaction <*date*, *city*> may conduct selection on the field level considering the selection efficiency. Hence, the "hybrid" refers to the different selection granularity between feature interactions.
>
> ### **W3: Writing**
> We appreciate the reviewer pointing out the confusing and unclear points in our writing. Accordingly, we address each point the reviewer mentioned as follows.
>
> #### *Introduction*
> Please kindly allow us to highlight our papers' contributions here. We will revise the last paragraph of the introduction section accordingly.
> First and foremost, we extend the granularity of feature interaction selections from field level to value level. To the best of our knowledge, we are the first to propose such an extension.
> Second, to tackle the size explosion problem brought by such an extension, we decompose the selection space via tensor factorization.
> Third, for the sake of selection efficiency, we propose a hybrid-grained selection approach (hybrid between value level and field level) named OptFeature.
> Finally, we conduct extensive experiments to validate the efficiency and effectiveness of OptFeature on three large-scale benchmark datasets. Multiple ablation studies are also conducted to investigate different aspects of our approach, which can help readers better understand and utilize it in their works.
>
> #### *Related Work (Section 2.1)*
> We thank the reviewer for pointing this out. Generally speaking, the neural architecture search techniques are part of the techniques that inspired several previous works on feature interaction selection[1,2]. This is the same as continuous sparsification[3] in Section 2.2. To clarify the reviewer's concern, we will merge these two sections and highlight their relationship with our work at the beginning.
>
> #### *Section 3.3.2*
> We explicitly want to thank the reviewer for pointing this out, which helps to make our paper more self-contained. We forget to mention the determination of $\alpha$ in our paper. The reason is that determining $\alpha$ makes no difference from DARTS[4] and previous work[5,6]. It is relaxed as a continuous vector (each element between 0 and 1) during searching and discretized as a deterministic vector (each element is either 0 or 1). Please check the following for determining $\alpha$ in Section 3.3.2.
> - Inspired by DARTS[4] and its success in previous works[1,5,6], we relax the hybrid tensor $\alpha$ into a continuous tensor $\alpha_c \in \mathcal{R}^{n^t}$, which can be trained jointly with other parameters via gradient descent. To ensure convergence, we apply the sigmoid function over $\alpha_c$. Hence, during training time, Eq 6 (original Eq 6 $\mathbf{A} = \alpha \mathbf{A}_f + (1 - \alpha) \mathbf{A}_v$) can be rewritten as:
> $$\mathbf{A} = \text{sigmoid}(\alpha_c) \mathbf{A}_f + (1 - \text{sigmoid}(\alpha_c)) \mathbf{A}_v$$
>
> We will also add the following sentence to Section 3.3.3, which describes how we conduct retraining.
> - During the re-training stage, the optimal hybrid tensor is determined as $\alpha^* = \mathbb{1}_{\alpha_c > 0}$ following previous works[4-6].
>
>
>
> ### **Reference**
> 1. Autofis: Automatic feature interaction selection in factorization models for click-through rate prediction (KDD2020)
> 2. Autoias: Automatic integrated architecture searcher for click-through rate prediction (CIKM 2021)
> 3. Progressive feature interaction search for deep sparse network (NeurIPS 2021)
> 4. DARTS: Differentiable Architecture Search (ICLR 2019)
> 5. Memorize, factorize, or be naive: Learning optimal feature interaction methods for CTR prediction (ICDE 2022)
> 6. GAIN: A Gated Adaptive Feature Interaction Network for Click-Through Rate Prediction (Sensors 2022)
>
> *Please notice that we use the format paper "title(venue)" here due to the page limit. We fully appreciate all author's contributions to the community.*

---

> > ### Author Response · Authors · 2023-08-15
> >
> > Hi Reviewer Bzgv, We noticed that you increased your rating from 3 to 4, but we didn't receive your response. May we ask if there are any further concerns? Kindly let us know if you have any questions or any unaddressed concerns.

---

> > > ### Comment · Reviewer_Bzgv · 2023-08-18
> > >
> > > I have read all the reviews and the rebuttals, and I raised my rating from 3 to 4.

---

> > > > ### Author Response · Authors · 2023-08-18
> > > >
> > > > Hi Review Bzgv, Thanks for the response. We will fix all your concerns as discussed in the later version. Given that 4 is still a negative score, could you kindly inform us if any of your concerns have not been properly addressed? We would happily discuss them with you, as constructive criticism is valuable for improving the quality of research papers. Thanks!

---

> > > > > ### Author Response · Authors · 2023-08-20
> > > > >
> > > > > Again, thank you so much for carefully reading our response and for the assessment based on it.
> > > > >
> > > > > The rebuttal phase is drawing to a close. If possible, we would appreciate any additional feedback for further improving the quality of our paper. We believe that we have addressed all the concerns and weaknesses raised in your initial review, but the score remains borderline rejection. Therefore, we conjecture that you might still have some remaining concerns regarding our paper, which would be very useful for further improvement in the revision.
> > > > >
> > > > > We hope to be able to work together with you to arrive at a consistent decision about the acceptability of our papers.

---

> ### Author Response · Authors · 2023-08-18
>
> Hi Reviewer Bzgv,
> We would again appreciate the valuable and thoughtful review. Since the deadline for the discussion period is approaching, it would be great to have feedback on if our response addresses the concerns raised in your initial review.

---

### Official Review · Reviewer_57Bi · 2023-07-06

**Soundness:** 3 good
**Presentation:** 4 excellent
**Contribution:** 3 good
**Rating:** 6
**Confidence:** 3

**Summary:**

This paper tackles the problem of modeling fine-grained feature interactions in high-dimensional sparse features.
A hybrid-grained feature interaction selection method is proposed, which operates on both field and value for deep sparse networks.
To handle the increase in computation, a decomposed form of the selection space is done, which greatly reduces the computational requirements of modeling.
Results on deep sparse networks benchmarks show that the proposed method achieves SOTA results while being more computationally efficient.

**Strengths:**

- Strong results in terms of performance on established benchmarks and computational efficiency, demonstrating the effectiveness of the proposed method.
- The proposed method seems generalizable and can be applied to other methods.
- All experimental parameters are provided, making reproduction straightforward.
- The writing is fairly clear and easy to understand.

**Weaknesses:**

- Experimental results:
  - The proposed method is a simple tensor decomposition for improved efficiency and the additional consideration of more features. Such choices are generalizable to other architectures (as mentioned in lines 109-112) but this is not demonstrated in the paper. I would like to see the application of the proposed components to other existing approaches.
- Significance of results:
  - The AUC and Logloss scores differ by less than 0.001 between the proposed method and the previous SOTA. Is this significant? I suggest the authors add confidence intervals to Table 1 and 2 for easier comparison.

**Questions:**

See weaknesses.

**Limitations:**

The limitations are discussed at the end of the paper.

---

> ### Author Rebuttal · Authors · 2023-08-10
>
> Hi Reviewer 57Bi:
>
> Thanks for your effort in reviewing our paper and appreciating our effort. Below are responses to your questions.
>
> ### **W1: generalization**
> We deeply agree with the reviewer about the importance of generalization.
>
> To investigate this aspect, we include ablation studies regarding various interaction operations, embedding dimensions and feature interaction orders in Appendix C1, C2 and C3, respectively. However, the reviewer's comment reminds us to be self-contained in the main text. We will append additional references to the appendix sections in the main text regarding the generalization issue. Hopefully, this can address the reviewer's concern.
>
> More broadly speaking, we agree with the reviewer that it would be interesting to see how our method can be combined with more complex methods, such as embedding dimension search or feature interaction operation selection. However, the major difficulty lies in disentangling the influence between different methods, which may lead to sub-optimal results.  This, frankly speaking, is one of our future research projects. Kindly pay attention to our future papers regarding this aspect.
>
> ### **W2: significance**
> Thanks for pointing this out! Adding the statistical analysis can better validate our method. Please check the common weaknesses(Cw1) for our response.

---

> > ### Comment · Reviewer_57Bi · 2023-08-16
> > **Follow-up by Reviewer**
> >
> > Thank you for providing detailed responses to my concerns. I have read through all the other reviews and responses. I will maintain my rating of "Weak Accept".

---

> > > ### Author Response · Authors · 2023-08-17
> > >
> > > Thanks again for appreciating our effort. We will make the corresponding change to the paper later.

---

### Official Review · Reviewer_21ZP · 2023-07-07

**Soundness:** 3 good
**Presentation:** 3 good
**Contribution:** 2 fair
**Rating:** 6
**Confidence:** 4

**Summary:**

This paper introduces a hybrid-grained feature interaction selection approach that targets both feature field and feature value for deep sparse networks and decomposes the selection space using tensor factorization and calculating the corresponding parameters on the fly.


**Strengths:**

Extending the selection granularity of feature interactions from the field to the value level.

Introduce a hybrid-grained feature interaction selection space, which explicitly considers the relation between field-level and value-level.

The tensor decomposition and the sparsification are combined to perform selection on the shrinking space.


**Weaknesses:**

1.The evaluation datasets are pretty small (the feature number is around 11-26). For recommendation systems optimization work, it is usually better to show the results in large-scale datasets like industrial datasets to demonstrate the scalability and performance.


2.Missing several references:

AutoFAS，

NAS-CTR，

AutoIAS,

GAIN: A Gated Adaptive Feature Interaction Network for Click-Through Rate Prediction

Maybe adding some discussions or comparisons to them is better.





**Questions:**

See weakness

**Limitations:**

Yes, the authors discussed some limitations.

---

> ### Author Rebuttal · Authors · 2023-08-10
>
> Hi Reviewer 21ZP:
>
> Thanks for your effort in reviewing our paper and offering constructive suggestions.
>
> ### **W1: small datasets**
> We thank the reviewer's helpful suggestion in extending datasets with relatively different statistics. Our response to your concern is split into the following two parts.
>
> First, to ensure we are on the same page, we want to highlight the concepts of *feature field* and *feature value*. We include the following table regarding the statistics of our datasets (also detailedly described in Appendix B1). As we can observe, the number of feature values, which influence the maximum selection space, is around $\sim 10^6$ level.
>
> | Dataset | \#Samples | \#field | \#values | pos ratio |
> | --- | --- | --- | --- | --- |
> | Criteo | $4.6 \times 10^7$ | 39 | $6.8 \times 10^6$ | 0.2562
> | Avazu  | $4.0 \times 10^7$ | 24 | $4.4 \times 10^6$ | 0.1698
> | KDD12  | $1.5 \times 10^8$ | 11 | $6.0 \times 10^6$ | 0.0445
>
> Second, we add one additional experiment on a private large-scale industrial dataset. This dataset contains nearly 3.5 million samples with 134 feature fields and 2498 feature values. Please notice that this dataset has more feature fields and fewer feature values, which differs from the previous benchmarks with fewer feature fields and larger feature values. The following results can further prove the effectiveness of OptFeature. The observations are also consistent with those on the other three public benchmarks. We will include these results in the appendix later.
>
> | |  AUC | LogLoss |
> | --- | --- | --- |
> | LR         | 0.7745 | 0.2189 |
> | FM         | 0.7780 | 0.2181 |
> | FNN        | 0.7838 | 0.2168 |
> | DeepFM     | 0.7824 | 0.2179 |
> | DCN        | 0.7844 | 0.2167 |
> | IPNN       | 0.7883 | 0.2147 |
> | AutoFIS    | 0.7889 | 0.2146 |
> | PROFIT     | 0.7849 | 0.2161 |
> | OptFeature | 0.7893 | 0.2142 |
>
>
> ### **W2: missing reference**
> Thanks for referring to these relevant papers, as they are relevant and related. Please refer to the common response(CW2) for our response and corresponding changes.

---

> > ### Comment · Reviewer_21ZP · 2023-08-10
> > **Thanks for providing the results on large-scale industrial dataset.**
> >
> > Thanks for providing the results on the large-scale industrial dataset. I think the new comparisons addressed my concerns. I increase my score from borderline accept to weak accept.

---

> > > ### Author Response · Authors · 2023-08-11
> > >
> > > We sincerely thank reviewer 21ZP for the further feedback, and we are glad that your concerns are addressed. We will make the proper change in the paper later.

---

### Author Rebuttal · Authors · 2023-08-10

Hi Reviewers and PCs:

We want to thank all your effort in helping us improve this paper. Below are some of the common concerns. Kindly notice that we try to use points to answer reviewers' questions, as some of the weaknesses and questions are repetitive or similar.

### **CW1: result significance**
Thanks to all reviewers for highlighting this point. Here we list a table summarizing the recent and relevant papers working on the datasets we adopted. We list their relative improvements compared with the highest baseline in the original paper(N/A means the corresponding result is not contained). Compared with our relative improvement(last line of the table), We can observe that the improvement achieved by our proposed method makes sense over all three datasets.

| | Criteo AUC | Criteo LogLoss | Avazu AUC | Avazu LogLoss | KDD12 AUC | KDD12 LogLoss |
| ---- | ---- | ---- | ---- | ---- | ---- | ---- |
| AutoInt[1]        | +0.0052 | +0.0053 | -0.0006 | +0.0005 | +0.0084 | +0.0020 |
| AutoFIS[2]        | +0.0001 | +0.0000 | +0.0016 | +0.0009 |   N/A   |   N/A   |
| PROFIT[3]         | +0.0001 | +0.0001 | +0.0027 | +0.0046 |   N/A   |   N/A   |
| AutoIAS[4]        | -0.0045 | -0.0943 | -0.0001 | -0.0003 |   N/A   |   N/A   |
| GAIN[5]           | +0.0005 | +0.0004 | +0.0001 | +0.0001 |   N/A   |   N/A   |
| NAS-CTR[6]        | +0.0009 | +0.0004 | +0.0073 | +0.0049 |   N/A   |   N/A   |
| OptFeature(ours)  | +0.0004 | +0.0004 | +0.0019 | +0.0005 | +0.0013 | +0.0002 |

We also conduct a two-tailed t-test over our OptFeature and the best-performed baseline, the corresponding $p$-value $<0.005$. This is usually denoted as statistically significant[2,6]. Corresponding changes will be made to the paper later.

### **CW2: missing discussion of some related works** and **unclear expression of novelty**

We thank reviewers 21ZP, Bzgv and eeCX for pointing out our flaws. We revise the 2nd paragraph in Section 2.3 to address these two issues. Kindly check the following:

```
With the advancement of neural architecture search[7-9] and continuous sparsification[10], various methods have been proposed to select the informative feature interactions[2,3] and reduce computational costs. AutoFis[2] employs a sparse optimizer to select suitable feature interactions at the field level. PROFIT[3] formulates field-level feature interaction selection within a distilled search space. It employs a progressive search for efficient exploration. AutoIAS[4] takes one step further to integrate the feature interaction selection as part of the search space and jointly conduct the search with other components like MLP architecture or embedding dimension. GAIN[5], on the other hand, focuses on the DCN[11]-like architectures and conducts the feature interaction selection jointly with the model training. However, all previous works conduct feature interaction selection on the field level. Our work builds on the existing approaches for modelling feature interactions in DSNs. More precisely, We extend the selection granularity to the value level and propose a hybrid-grained selection approach.

Furthermore, there also exists some works such as OptInter[12], AutoFeature[13], and NAS-CTR[6] take a different angle and search for suitable operations (such as inner product, outer product or element-wise sum) to model each feature interaction properly. These works are perpendicular to our study.
```

### **Reference**
1. Autoint: Automatic feature interaction learning via self-attentive neural networks (CIKM 2019)
2. Autofis: Automatic feature interaction selection in factorization models for click-through rate prediction (KDD2020)
3. Progressive feature interaction search for deep sparse network (NeurIPS 2021)
4. Autoias: Automatic integrated architecture searcher for click-through rate prediction (CIKM 2021)
5. GAIN: A Gated Adaptive Feature Interaction Network for Click-Through Rate Prediction (Sensors 2022)
6. NAS-CTR: Efficient Neural Architecture Search for Click-Through Rate Prediction (SIGIR 2022)
7. Neural Architecture Search with Reinforcement Learning (ICLR 2017)
8. Neural architecture optimization (NeurIPS 2018)
9. DARTS: Differentiable Architecture Search (ICLR 2019)
10. Estimating or propagating gradients through stochastic neurons for conditional computation (CoRR 2013)
11. Deep & cross network for ad click predictions (ADKDD@KDD 2017)
12. Memorize, factorize, or be naive: Learning optimal feature interaction methods for CTR prediction (ICDE 2022)
13. Autofeature: Searching for feature interactions and their architectures for click-through rate prediction (CIKM 2020)

*Please notice that we use the format paper "title(venue)" here due to the page limit. We fully appreciate all author's contributions to the community.*

---

### Decision · Program_Chairs · 2023-09-21

**Decision:**

Accept (poster)

**Comment:**

After a quite active discussion period, all the concerns of the reviewers are addressed and the promised paper revision is doable for before the camera deadline. Therefore, the ACs recommand accepting the paper.